# FAST AUTOREGRESSIVE VIDEO GENERATION WITH DIAGONAL DECODING

## ABSTRACT

Autoregressive Transformers demonstrate impressive performance in generation models. However, their sequential, token-by-token decoding becomes a severe bottleneck for video generation, which may require generating tens of thousands of tokens sequentially. In this paper, we introduce Diagonal Decoding (DiagD), a training-free inference acceleration algorithm that exploits spatiotemporal correlations to speed up autoregressively pre-trained models. DiagD generates tokens simultaneously along diagonal trajectories in the spatial-temporal token grid, enabling parallel decoding within frames and partial overlap across successive frames. The proposed algorithm is versatile and adaptive to various generative models and tasks and offers adjustable trade-offs between speed and visual quality. Furthermore, we propose a cost-effective fine-tuning strategy that aligns the attention patterns of the model with the new decoding order to demonstrate the potential of training with DiagD. Experiments on several autoregressive video generation models and datasets demonstrate that DiagD achieves up to $10\times$ speed-up over naive sequential decoding, while preserving comparable visual fidelity.

## 1 INTRODUCTION

Recent advances in video generation models have achieved a significant level of performance in both diffusion (Lin et al., 2024; Yang et al., 2024; Xu et al., 2024) and autoregressive (Kondratyuk et al., 2023; Agarwal et al., 2025; Kanervisto et al., 2025) based methods. These models demonstrate impressive capabilities in learning foundational knowledge from raw videos and generating high-fidelity, controllable video outputs (Brooks et al., 2024). Consequently, video generation models have also been adopted in various domains in AI such as world modeling (Ha & Schmidhuber, 2018; Agarwal et al., 2025; Kanervisto et al., 2025) and embodied AI (Yang et al., 2023), illustrating their potential power to serve as digital twins of the real world.

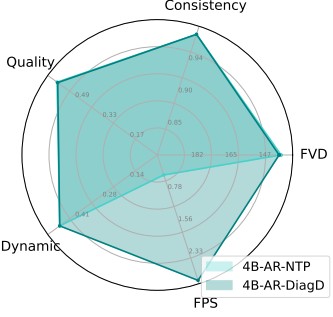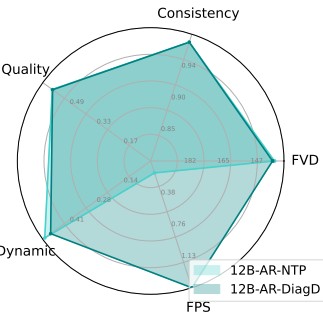

Figure 1: Comparisons between naive Next-Token Prediction (NTP) and the proposed Diagonal Decoding (DiagD) on Cosmos (Agarwal et al., 2025) autoregressive models. DiagD achieves 6 to $10\times$ speedup with negligible degradation on visual quality among different scales of models.

Compared with diffusion models, autoregressive Transformers exhibit unique features as shown by the blooming of Large Language Models (LLMs) (Radford et al., 2019; Brown et al., 2020) in recent years, including zero-shot emergent in-context learning capabilities (Zhang et al., 2025), and scaling laws (Kaplan et al., 2020; Pearce et al., 2024). Leveraging architectures similar to LLMs enables

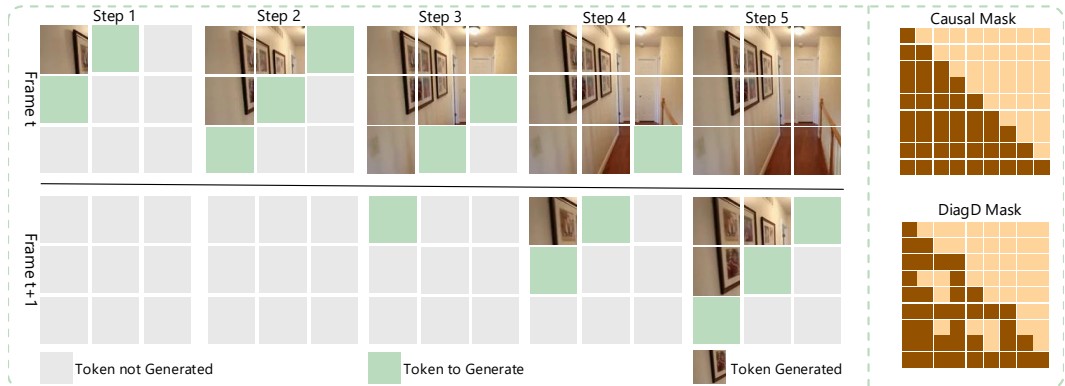

Figure 2: Left: An illustration of the proposed Diagonal Decoding algorithm with $d = 3$ and $k = 1$. Spatially, tokens along the same diagonal within each frame are generated in parallel. Temporally, our method generates the top-left tokens of the subsequent frame before completing the current frame. Right: An illustration of the causal attention mask and the DiagD mask utilized in fine-tuning.

vision models to inherit these advancements and naturally extend to multi-modal inputs. Additionally, autoregressive models can generate videos of arbitrary length in a streaming paradigm, which is challenging for most diffusion models.

However, video generation models usually utilize a visual tokenizer (Esser et al., 2021; Tang et al., 2024) to transform raw videos to tens of thousands of tokens, which poses a significant bottleneck for autoregressive models that generate tokens sequentially, especially when generating high-resolution, long-duration videos. The bottleneck can be divided into three main challenges. Firstly, the naive next-token prediction mechanism leaves computational resources underutilized and thus leads to slow and costly generation. Secondly, previous visual autoregressive models (Yu et al., 2023b; Kondratyuk et al., 2023; Bai et al., 2024) generate tokens following a fixed raster-scan order (i.e., left-to-right, top-to-bottom, frame-by-frame), which creates suboptimal generation trajectories for image and video synthesis. Thirdly, the paradigm of autoregressive video generation remains underexplored.

In this paper, we propose **Diagonal Decoding (DiagD)**, an algorithm that utilizes redundant information in video representations by generating diagonal tokens in both spatial and temporal adjacent regions simultaneously. As illustrated in Figure 2, instead of generating tokens sequentially in a raster-scan order, our method generates the diagonals of images from top left to bottom right, with tokens along the same diagonal produced in parallel at each step. By stacking frames together, diagonal decoding can be seamlessly applied to video generation. Notably, our method is training-free and functions as a plug-and-play module on autoregressively pretrained models, requiring only $5\%$ of the generation steps and up to a $10\times$ speedup in inference latency with negligible quality degradation, compared with next token prediction. We introduce hyperparameters to control the acceleration ratio in both spatial and temporal dimensions, allowing flexible adjustments to the trade-off between speed and performance. Additionally, we also provide a fine-tuning strategy to demonstrate that training with DiagD brings further improvements in performance.

We evaluate the performance and generalizability of Diagonal Decoding across various autoregressive video generation models, tasks, and datasets. Specifically, as shown in Figure 1, on the Cosmos (Agarwal et al., 2025) world model, our method accelerates the inference by $10+$ times while achieving similar visual quality to next token prediction on tasks including video continuation and text-guided video generation. On WHAM (Kanervisto et al., 2025), a world model for games that produces multi-modal outputs, our spatial-only acceleration variant achieves approximately $4\times$ speedups while preserving generation quality. In addition, we also train autoregressive Transformer models from scratch to validate the performance of our method on different scales of models.

In summary, our contributions are threefold:

**i)** We propose Diagonal decoding, a plug-and-play acceleration algorithm for autoregressive video generation, achieving up to $10\times$ speedup in inference while maintaining generation quality.

**ii)** The proposed decoding algorithm demonstrates strong generalization capability across various autoregressive implementations including arbitrary visual tokenizers (with or without temporal compression), arbitrary resolutions, and diverse generation tasks.

**iii)** Besides inference, fine-tuning with diagonal decoding consistently improves the model performance, which provides inspiration for training video generation models in the future work.

## 2 RELATED WORK

**Video Generative Models** Video generative models have advanced rapidly in recent years, achieving impressive results in producing long-range, high-fidelity and controllable videos (Ho et al., 2022; Kondratyuk et al., 2023; Yu et al., 2023c; Brooks et al., 2024; Liu et al., 2024b; Ma et al., 2024a;b). Most video generation models consist of two components: a visual tokenizer (Li et al., 2024; Tang et al., 2024; Wang et al., 2024) that converts raw images and videos into latent representations, and a generative model that synthesizes these latents. However, representing a video clip requires a large number of latents. For instance, a 16-frame video can produce between 40k and 160k tokens. The large number of latents creates a significant computational bottleneck for the generative model.

For the paradigm, diffusion (Ho et al., 2020) and autoregressive generation (Vaswani et al., 2017) are the two most popular approaches in video generation. In this paper, we focus on autoregressive Transformers, as they demonstrate performance on par with diffusion models (Kondratyuk et al., 2023; Yu et al., 2023c), while inheriting key advantages from large language models, such as zero-shot in-context learning (Zhang et al., 2025), long-range generation capabilities (Liu et al., 2024a), and the ability to smoothly integrate multiple modalities (Kondratyuk et al., 2023).

**Parallel Decoding in Generative Models** Parallel decoding has been widely explored in Transformer-based models to accelerate inference. Inspired by masked language models (Ghazvininejad et al., 2019; Guo et al., 2020), MaskGIT (Chang et al., 2022) and MAGVIT (Yu et al., 2023a) introduces masked generative Transformers that generate tokens in parallel through iterative denoising. LFormer (Li et al., 2023) divides tokens into several L-shaped blocks in images, and generates tokens in each block in parallel. However, this method requires retraining the model. Recently, ZipAR (He et al., 2024) proposes a parallel decoding algorithm for image generation by exploiting local token dependencies. Different from previous works, our diagonal decoding method is training free, operates at the video level, and achieves greater speedup ratios by handling temporal dependencies directly.

## 3 METHOD

### 3.1 BACKGROUND

In this section, we introduce the proposed Diagonal Decoding method. We begin with an overview of the task. Given a raw video $\mathbf{x}$ composed of a sequence of frames, a discrete VAE is used to encode the frames into a sequence of discrete tokens $\mathbf{c}$:

$$
\begin{aligned}
\mathbf{x} &= (x_1, \cdots, x_T), \\
\mathbf{c} &= (c_1, \cdots, c_n, c_{n+1}, \cdots, c_{2n}, c_{2n+1}, \cdots c_N),
\end{aligned}
\tag{1}
$$

where $T$ denotes the number of frames, $n$ denotes the number of tokens to represent each frame, and $N = T \cdot n$ denotes the total number of encoded tokens. The autoregressive Transformer processes this sequence and learns to model the spatial and temporal dynamics of the video through next-token prediction. The training objective is to maximize the joint probability of each token, where the model predicts the current token based on all previously generated tokens:

$$
\max_{\theta} \quad p_{\theta}(\mathbf{c}) = \prod_{i=1}^{N} p_{\theta}\left(c_i \mid c_1, c_2, \cdots, c_{i-1}\right),
\tag{2}
$$

where $p_{\theta}$ denotes the Transformer model parameterized by $\theta$. During inference, the model generates tokens sequentially through next-token prediction, which is identical to a raster-scan order once the sequence is reshaped back into a 2D structure. Finally, the decoder of the discrete VAE reconstructs the predicted tokens into videos in the RGB space.

### 3.2 DIAGONAL DECODING

The motivation of our method arises from intuitive observations on consecutive frames in a video, which can be summarized in two key insights. As shown in Figure 2, the first insight is that patches

exhibit stronger correlations with their spatial neighbors than with sequential ones. For example, the first patch in each row is more related to the first patch in the previous row than to the last patch in the same row, despite the latter being its sequential predecessor. Secondly, due to the temporal redundancy of videos, patches from consecutive frames that occupy similar relative positions are highly likely to be similar to each other. We empirically validate our observations in Figure 6. As a result, we find that sequential autoregressive generation is not only counterintuitive but also inefficient, and we propose leveraging these spatial and temporal correlations to accelerate the generation process.

Specifically, we propose Diagonal Decoding, an iterative algorithm that generates tokens along diagonal paths in the spatial-temporal token grid. Spatially, within each frame, tokens along the same diagonal are generated in parallel, leveraging the strong local dependencies between neighboring patches. And temporally, as illustrated in Figure 2, by stacking frames together, our method generates the top-left tokens of the next frame before completing the current frame, as these tokens are less likely to depend on the bottom-right tokens that have not yet been generated.

Formally, let $h$ and $w$ denote the height and width of a frame, respectively, and let $c_{t,i,j}$ represent the token corresponding to the patch in row $i$ and column $j$ of the $t$-th frame in the video. We introduce two hyperparameters to define our algorithm. The parameter $k$ represents the number of existing spatial neighbors in the previous row available when generating the current token. In other words, $c_{t,i,j}$ is generated after all tokens satisfying $c_{t,\leq i,\leq j+k}$ have been generated. Then we can calculate the number of iterations to generate all tokens in one frame:

$$s_{\text{spa}} = (h-1) \cdot k + w. \tag{3}$$

Compared to the standard next-token prediction, which requires $h \cdot w$ steps, the acceleration ratio of spatial diagonal decoding is given by:

$$r_{\text{spa}} = \frac{h \cdot w}{(h-1) \cdot k + w} \approx \frac{\min\{h, w\}}{k+1}. \tag{4}$$

From the temporal aspect, we introduce temporal delay $d$ to represent the number of diagonal lines that must be generated in the previous frame before starting the next frame. The value of $d$ ranges in $[1, s_{\text{spa}}]$, where $d = 1$ represents the extreme case where the next frame begins generating immediately after the first token of the previous frame is produced, and $d = s_{\text{spa}}$ corresponds to no temporal level acceleration being utilized. Equipped with both spatial and temporal diagonal decoding, the number of iterations to generate all tokens in $T$ frames can be written as:

$$s_{\text{tep}} = (T-1) \cdot d + s_{\text{spa}}. \tag{5}$$

As illustrated in Figure 2, we set $k = 1$ and $d = k \cdot h$ to balance generation quality and speed in most of our experiments, while naturally aligning spatial and temporal acceleration within a unified diagonal decoding framework. The total speedup ratio compared to next-token prediction is given by:

$$r_{\text{diag}} = \frac{T \cdot h \cdot w}{(T-1) \cdot h + h + w - 1} \approx w. \tag{6}$$

The derivation of Equation (4) and (6) are shown in the appendix H. This indicates that the acceleration is roughly proportional to the width of the video resolution, significantly reducing the number of decoding iterations relative to standard autoregressive methods. We provide analysis on both hyperparameters $k$ and $d$ in experiments.

**Discussion** The $k$ and $d$ hyperparameters, flexibly control the tradeoff between inference speed and generation quality, enhancing the versatility of DiagD. For pure video generation models (e.g., Cosmos (Agarwal et al., 2025)), both spatial and temporal acceleration can be enabled to achieve the highest inference speed. On the other hand, for models with multimodal outputs (e.g., WHAM (Kanervisto et al., 2025), which generates paired images and actions in games), where temporal acceleration is not applicable, setting $d = s_{\text{spa}}$ allows the model to leverage spatial diagonal decoding alone.

The spatial-only variant of our diagonal decoding shares insights with ZipAR (He et al., 2024), a decoding algorithm for text-to-image generation, but introduces a key innovation: leveraging temporal redundancies across frames to enhance efficiency even for the spatial-only algorithm. Specifically, implementing diagonal decoding introduces a training-inference gap, as the first token in each row is conditioned on the last generated token of the previous row during training, and such dependency is

absent during inference. Previous methods use the last token in the previous row, $c_{t,i-1,j+k}$, as the predecessor for generating token $c_{t,i,j}$. In contrast, our approach leverages temporal information by using $c_{t-1,i,j}$, the token at the same position in the previous frame, which provides additional context and enables more accurate predictions. As a result, while ZipAR requires a large $k = 16$ to maintain visual quality, our spatial-only diagonal decoding achieves higher speedups (e.g., $k = 1$ on large scale models or 2 on smaller ones) without compromising visual fidelity.

## 3.3 FINE-TUNING WITH DIAGD ATTENTION MASK

Diagonal decoding introduces a training–inference gap for autoregressively pre-trained models, as it deviates from the standard raster-scan generation order. Despite this mismatch, our experiments demonstrate that models are able to neglect the gap and achieve substantial speedups with minimal performance loss in a training-free setting. Nonetheless, we hypothesize that directly training with diagonal decoding (DiagD) can further enhance both model performance and efficiency. Here we present a fine-tuning strategy using DiagD and leave full training from scratch as future work.

As illustrated in Figure 2, we adapt the model by replacing the standard causal attention mask with a DiagD-aligned attention mask. This modification ensures that each token attends only to its DiagD predecessors, rather than to raster-ordered positions, thereby reducing the discrepancy between training and inference. Empirically, we find that fine-tuning the model for merely 1k steps is sufficient to close most of the performance gap, validating the practicality and effectiveness of this approach.

## 4 EXPERIMENTS

In this section, we present experimental results on the proposed Diagonal Decoding algorithm. We start with experimental setups described in Section 4.1, followed by the main results on various models illustrated in Section 4.2. We provide analysis and case studies in Section 4.3 and Section 4.4.

### 4.1 SETUPS

#### 4.1.1 BASELINES

We consider three representative models as our baselines to validate the performance of Diagonal Decoding with temporal and spatial accelerations, respectively. To study the relations between model scales and the performance of DiagD, we also train autoregressive models from scratch.

**Cosmos** Cosmos (Agarwal et al., 2025) is a world foundation model collection that integrates multiple pre-trained models. We utilize the autoregressive models, which are equipped with a discrete video tokenizer that provides $8\times$ temporal compression and $16\times$ on spatial. As a result, for 8 frames with $640 \times 1024$ as the raw resolution, it is encoded into latent discrete tokens with size $40 \times 64 = 2,560$, i.e., $h = 40$ and $w = 64$ following our notations. Experiments on Cosmos demonstrate the generalizability of Diagonal Decoding on representations with temporal compressions.

**WHAM** The World and Human Action Model (WHAM) (Kanervisto et al., 2025) is an autoregressive generative model on the game environment, which is capable of generating accurate and coherent game scenes following instructions from users. Different from Cosmos which produces videos solely, WHAM takes interleaved concatenations of images and actions as input and output, to receive controls and generate consequences. Therefore, WHAM utilizes an image tokenizer with $10\times$ spatial compression only, which transforms a raw game scene with $180 \times 300$ into $18 \times 30 = 540$ tokens, with $h = 18$ and $w = 30$ as a result. We test DiagD with spatial acceleration on WHAM, considering that the action will be given by user after the previous game scene has been generated.

**MC-AR** To study the performance of Diagonal Decoding on different scales of models, as well as validating the proposed fine-tuning strategy, we train a series of models from scratch. Specifically, we utilize the VPT dataset (Baker et al., 2022) which consists of pairs of game scenes and actions on the game Minecraft. We transform the raw game scenes with an image VQ-VAE (Patil et al., 2024) to latent tokens with size $14 \times 24 = 336$, i.e., $h = 14$ and $w = 24$. Then, a Transformer decoder is trained with next token prediction by taking the concatenation of game scenes and actions as input. We train model scales from 300M to 1.2B parameters. We leave detailed descriptions of baselines and the training procedure in the appendix G.

#### 4.1.2 EVALUATION SETUPS

**Metrics** For all models, we use one NVIDIA 80GB A100 GPU and batch size as 1 to obtain results. We propose separate metrics to assess the visual quality and inference speed. We follow common

Table 1: Quantitative evaluation of Cosmos on RealEstate10K (Zhou et al., 2018). "NTP" refers to the next-token prediction paradigm. DiagD $k = i$ $d = j$ denotes the Diagonal Decoding algorithm with different hyper-parameters. "STEP" refers to the number of forward passes required by the model to generate a video. "TP" represents throughput, i.e., number of tokens the autoregressive model can generate per second. Comparable numbers are underlined.

| Model | Algorithm | FVD↓ | Subject Cons.↑ | Image. Qual.↑ | Dynamic degree ↑ | FPS↑ | TP↑ | STEP $(k)$↓ |
|-------|-----------|------|----------------|---------------|------------------|------|-----|-------------|
| 14B | Diffusion | **129** | 0.976 | **0.677** | **0.61** | 0.08 | / | / |
| 4B | NTP | 136 | 0.978 | 0.604 | 0.49 | 0.38 | 120 | 7.68 |
|  | DiagD $k = 4$ $d = 20$ | 136 | **0.979** | 0.604 | 0.50 | 2.66 | 852 | 0.26 |
|  | DiagD $k = 2$ $d = 40$ | 137 | 0.978 | 0.600 | 0.49 | 3.02 | 966 | 0.30 |
|  | DiagD $k = 2$ $d = 10$ | 137 | 0.978 | 0.599 | 0.46 | **3.41** | **1097** | 0.15 |
| 12B | NTP | 135 | 0.978 | 0.602 | 0.54 | 0.15 | 49 | 7.68 |
|  | DiagD $k = 2$ $d = 40$ | 136 | 0.978 | 0.601 | 0.50 | 1.21 | 384 | 0.30 |
|  | DiagD $k = 2$ $d = 10$ | 136 | 0.978 | 0.600 | 0.51 | 1.50 | 480 | 0.19 |
|  | DiagD $k = 1$ $d = 40$ | 136 | 0.976 | 0.590 | 0.49 | 1.62 | 512 | 0.18 |
|  | DiagD $k = 1$ $d = 1$ | 139 | 0.967 | 0.564 | 0.51 | 1.71 | 549 | **0.11** |

practices and utilize metrics including Fréchet Video Distance (FVD) (Unterthiner et al., 2018) and VBench (Huang et al., 2024). We report Subject Consistency, Dynamic Degree, and Image Quality from VBench to comprehensively evaluate the generation performance. For the inference speed, we report three metrics. The Frames Per Second (FPS) generated by the model, calculated with the wall-clock time. Step denotes the number of forward passes for a model to generate the video. In addition, Throughput (TP) of output tokens per second is also recorded, which is a crucial metric in real-time applications.

In addition to automatic metrics, we also provide human evaluations, detailed in Section 4.4, to assess the generation results from various aspects including the general visual quality, object movement consistency, and the comparisons between two settings.

**Evaluation Dataset** For Cosmos, an open-sourced evaluation pipeline and dataset is absent. Therefore, we implement the pipeline by ourselves following details provided in their technical report (Agarwal et al., 2025). The validation dataset we selected spans autonomous driving (Yu et al., 2020), robotic manipulation (Walke et al., 2023), and camera trajectories (Zhou et al., 2018), which contains sufficiently diverse and rich dynamic motion information. For each dataset, we randomly sample 100 videos with 33 frames as the test set. For WHAM, we randomly select 100 videos with 100 frames from the official evaluation set due to the limitations on time and resource. Experiments on the whole test set with 1k videos will be reported in future revisions. For MC-AR, we split 100 video clips from VPT (Baker et al., 2022), each containing 16 frames.

## 4.2 MAIN RESULTS

In this section, we illustrate the main results of DiagD on various baseline models and tasks.

**Cosmos** We apply DiagD with temporal and spatial acceleration to Cosmos autoregressive models, with various combination of $d$ and $k$. We conduct experiments on different scales of Cosmos models on video continuation task. Conditioned on 9 initial frames, the model is required to generate the following 24 frames. We also show the performance of the 14B Diffusion model for reference.

We show representative results of DiagD variants with different $k$s and $d$s in Table 1 on RealEstate10K datasets, more results on robotic manipulation and driving scene are listed in appendix E. Compared to naive Next-Token Prediction (NTP), which requires 7.68k steps to generate 24 frames, DiagD reduces the step count to only 2% to 4%, enabling substantial parallelism in decoding. In terms of FPS measured by wall-clock time, DiagD achieves approximately $10\times$ speedup over NTP across various settings and model scales.

For visual quality, by adjusting $k$ and $d$, the variant ($k = 2, d = 10$) with $7\times$ speedup introduces negligible degradation in subject consistency, image quality, and dynamic degree, even in the 4B Cosmos model. For the 12B model, setting $k = 1, d = 1$ does not harm performance. Overall, DiagD significantly accelerates inference in a training-free manner with minimal impact on visual quality.

Table 2: Quantitative evaluation on WHAM. Each evaluation video has a duration of 10 seconds and a frame rate of 10 fps. For every video in this dataset, the initial ten frames along with the complete action sequence serve as prompts for generation. Second best results are underlined.

| WHAM | Algorithm | FVD↓ | Subject Cons.↑ | Image. Qual.↑ | Dynamic degree ↑ | FPS↑ | TP↑ | STEP ($k$)↓ |
|---|---|---|---|---|---|---|---|---|
| 200M | NTP | 367 | 0.747 | **0.628** | **0.983** | 0.23 | 124 | 54 |
| | DiagD $k=2$ | 462 | 0.723 | 0.613 | 0.967 | 0.75 | 405 | 6.4 |
| | DiagD $k=1$ | 472 | 0.727 | 0.624 | 0.902 | **0.97** | **524** | **4.7** |
| 1.6B | NTP | **336** | 0.747 | **0.628** | 0.975 | 0.12 | 65 | 54 |
| | DiagD $k=2$ | 378 | **0.749** | 0.627 | 0.975 | 0.34 | 184 | 6.4 |
| | DiagD $k=1$ | 365 | 0.739 | 0.625 | 0.975 | 0.40 | 216 | **4.7** |

**WHAM** On WHAM models, we validate the performance of DiagD with spatial acceleration solely. Specifically, we set $d = s_{spa}$ and $k$ to 1 or 2 in this setting. The generation task is challenging as the model is required to generate 100 frames conditioned on one initial frame and a sequence of actions, and slight errors in previous frames will cause huge performance drop due to error accumulation.

The results are shown in Table 2. The spatial only DiagD requires $10\%$ steps to generate a video compared to NTP, and brings 4 times speedups regarding FPS. This aligns with derivations in Equations (4) and (6), where temporal acceleration introduces an additional speedup of roughly $k+1$ times. In terms of visual quality, the larger 1.6B model exhibits less performance degradation compared to the 200M model, suggesting that larger models can better tolerate the training-inference gap introduced by diagonal decoding. Overall, these results demonstrate the effectiveness of the spatial variant of DiagD in balancing inference speed and visual fidelity across different model scales.

Table 3: Quantitative evaluation on 700M MC-AR. Fine-tuning with DiagD attention mask helps bridge the training-inference gap and improve performance. Second best numbers are underlined.

| Algorithm | FVD↓ | Subject Cons.↑ | Image. Qual.↑ | Dynamic degree ↑ | FPS↑ | TP↑ | STEP ($k$)↓ |
|---|---|---|---|---|---|---|---|
| NTP | **210** | **0.864** | **0.677** | 0.985 | 1.08 | 363 | 5.04 |
| DiagD w/o FT | 247 | 0.844 | 0.628 | 0.985 | 2.09 | 702 | 0.75 |
| DiagD w/ FT | 231 | 0.859 | 0.673 | **1.000** | 2.09 | 702 | 0.75 |

**MC-AR** We validate the proposed fine-tuning strategy on MC-AR models and analyze its scaling behavior in the appendix G. Specifically, we replace the standard causal attention mask in the pre-trained autoregressive Transformer with one aligned to DiagD, then fine-tune models for another 1k steps. As shown in Table 3, fine-tuning mitigates the training-inference gap, enhancing generation quality while preserving the fast inference speed of DiagD.

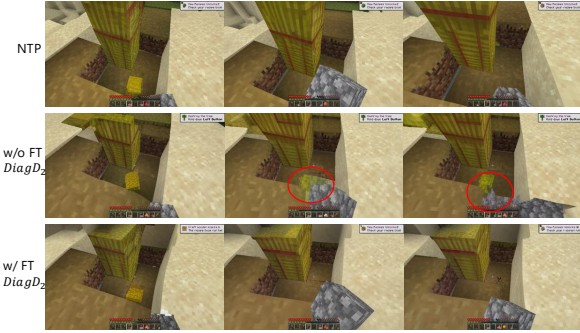

Figure 3: Qualitative comparison of results on MC-AR. Frames generated by models without fine-tuning may appear blurry, which can be mitigated with $1k$-step fine-tuning.

**Human Evaluation** We conduct human evaluations as a complement to automatic evaluations. For Cosmos-12B, we provide 10 videos generated by next-token prediction and DiagD, and ask participants to evaluate which one performs better in terms of the visual quality and camera consistency. For MC-AR, we ask participants to compare generation results from DiagD with or without fine-tuning, in terms of the visual quality and controllability. As shown in Figure 5, we find that: 1) DiagD and

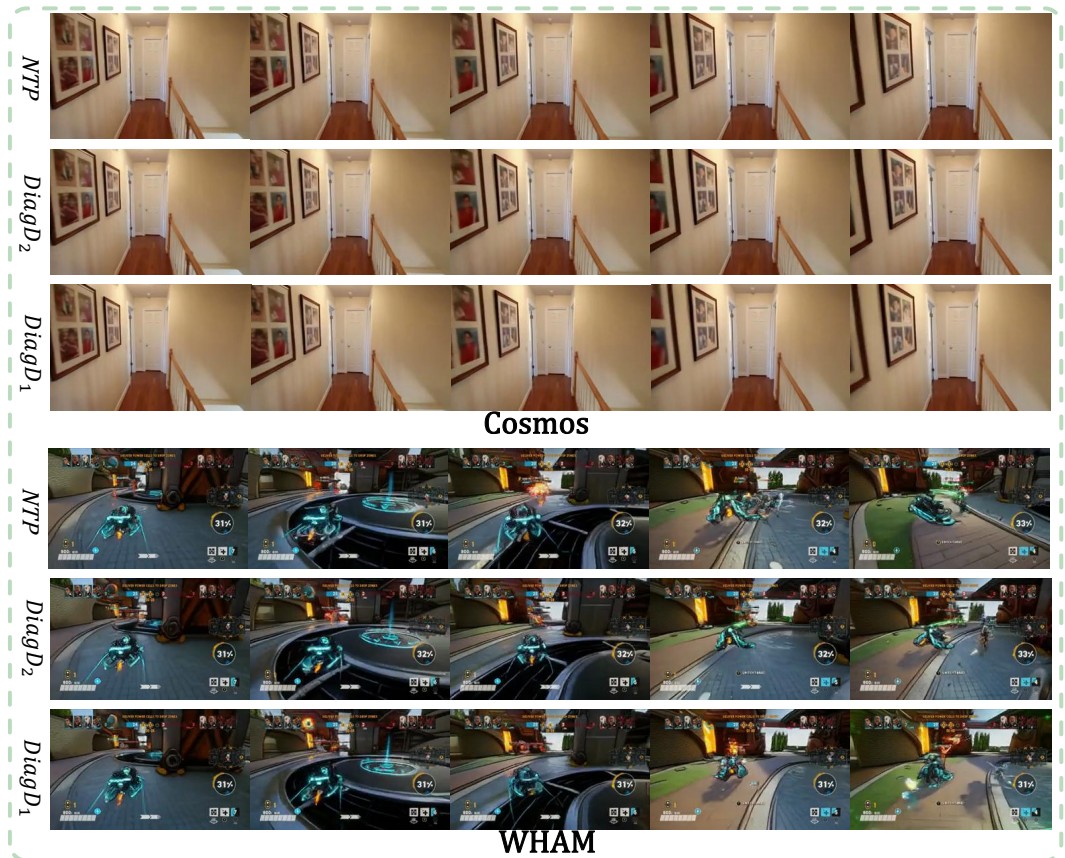

Figure 4: Qualitative analysis of Cosmos and WHAM. Videos generated by Cosmos-12B and 1.6B WHAM models using the next-token prediction paradigm (first row) and Diagonal Decoding under different configurations (bottom two rows) We uniformly sample 5 frames from 33-frames videos in Cosmos and 40-frames videos from those in WHAM.

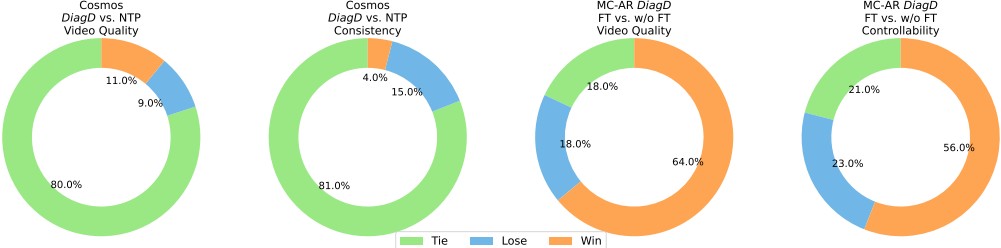

Figure 5: Human evaluation results for Cosmos-12B with DiagD and MC-AR 700M with or without DiagD fine-tuning. "Win" indicates the left setting outperforms the right one, while "Lose" represents the opposite. The results indicate that DiagD achieves similar performance to NTP, and fine-tuning helps it perform even better.

NTP generate videos with similar visual quality and semantic meaning; 2) fine-tuning help improve the visual quality significantly for a smaller 700M model.

## 4.3 ANALYSIS

**Attention Pattern** We visualize the attention map of the second frame generated by the Cosmos-Autoregressive-4B, as depicted in Figure 6. The diagonal patterns indicate that significant attention scores are allocated to tokens at fixed intervals, corresponding to tokens located in the same column of previous rows and the preceding frame. Spatially, tokens along the same diagonal exhibit notably high attention scores, indicating strong spatial relevance, as shown in the right square highlighted in the attention map. Temporally, tokens primarily attend to adjacent positions in the previous frame,

emphasizing temporal correlation, as illustrated by the left square in the attention map. The results of the attention map provide empirical support for the intuitive motivation introduced in Section 3.2.

**Scaling Effects** We also observe that DiagD achieves better performance and greater speedup on larger models, confirming that larger models capture more spatial and temporal properties in videos than smaller ones. Although Cosmos-4B and Cosmos-12B exhibit nearly identical FVD scores with next-token prediction, they demonstrate significantly different scores when using DiagD ($k = 1$). Additionally, we observed that both Cosmos-12B and WHAM-1.6B achieve higher FPS and superior visual quality with DiagD compared to their smaller counterparts employing next-

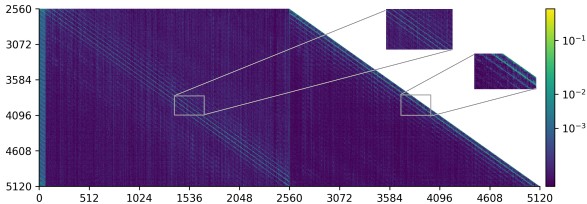

Figure 6: The attention scores of the second frame in the Cosmos-4B model are shown. The bright slash lines indicate that substantial attention scores are assigned to tokens at regular intervals, corresponding to those in temporally and spatially adjacent positions. The shown attention map is the mean value of all self-attention layers in the model.

token prediction. Therefore, DiagD may serve as an effective benchmark for evaluating whether models accurately capture spatial and temporal redundancies.

**Study on Hyperparameters** In Table 7 in the appendix, we present the results of various hyperparameter combinations for Cosmos-4B and Cosmos-12B. We found that $k$ has a more significant impact on controlling the speedup ratio than $d$. When $k$ values are similar, the FPS remains comparable. Additionally, increasing $d$ can enhance visual quality. By adjusting the values of $d$ and $k$, we can precisely balance visual quality and inference speed. This finding suggests that DiagD can be flexibly leveraged, eliminating the necessity of training smaller models solely for computational efficiency, decoupling the strong correlation between model size and inference speed.

## 4.4 CASE STUDY

In this section, we present case studies on the generation results of DiagD on different models. In the top half of Figure 4, we show results using DiagD on the Cosmos-12B model. Compared to results obtained by next-token prediction, DiagD provides consistent camera movements and comprehensive image details. The lower half of Figure shows results produced by the spatial variant DiagD with the 1.6B WHAM model. For game footage that features large motion and frequent scene changes, DiagD consistently delivers high-fidelity frames; in a 10-second, 100-frame sequence with pronounced motion, we observed very few error accumulation. Moreover, videos generated by standard next-token prediction and by DiagD exhibit almost identical object trajectories, demonstrating that the proposed algorithm preserves both controllability and visual quality over long-range generation.

In addition, we present cases to demonstrate the effectiveness of our fine-tuning method. As shown in Figure 3, fine-tuning for only 1k steps helps alleviate blurry areas, and provides videos with similar quality to that generated by NTP.

## 5 CONCLUSION

In this paper, we introduce Diagonal Decoding (DiagD), a training-free algorithm that significantly accelerates the inference speed of autoregressive video generation models. By leveraging spatial and temporal correlations in consecutive frames, DiagD generates tokens along diagonal paths, achieving substantial speedups while preserving visual fidelity. Through extensive experiments across diverse models, tasks, and datasets, we demonstrate the efficiency and generality of our approach, reducing the inference latency of Cosmos models by $10\times$ while maintaining their performance. Additionally, we propose a lightweight fine-tuning strategy to close the training-inference gap, further improving generation quality with minimal computational cost. As a result, DiagD provides a practical and scalable solution for real-time video generation, pushing the boundaries of what is achievable with autoregressive Transformers in downstream tasks and related applications.

Future research could focus on optimizing the training process to better adapt Diagonal Decoding for smaller models, enabling them to achieve higher acceleration ratios without sacrificing performance.

REPRODUCIBILITY

We provide comprehensive implementation details, including model architectures, configurations, and codes in Appendix and Supplementary materials to ensure reproducibility.

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

# ICLR Paper Appendix for Fast Autoregressive Video Generation with Diagonal Decoding

## A   Declaration of LLM Usage

We used large language models (LLMs), including ChatGPT, to support manuscript preparation. Their use was limited to language editing (grammar, spelling, and word choice), code formatting (e.g., adding comments to the code), and drafting figures to aid the creation of final visualizations. All scientific ideas, analyses, and conclusions were conceived, validated, and interpreted independently by the authors. We gratefully acknowledge the assistance of large language models in our work.

## B   Limitation

The Diagonal Decoding method introduced in this paper is heavily dependent on the capabilities of pre-trained models. Empirical observations and extensive experiments indicate that larger, more thoroughly trained models produce higher visual quality and exhibit lower cumulative errors. As noted in the main paper, smaller or less well-trained models often require larger values of $d$ and $k$ to maintain visual quality. For simpler inference tasks, such as video continuation, these smaller models tend to perform better compared to more complex tasks like conditional video generation.

## C    DIRECT COMPARISON WITH ZIPAR

We provide a straightforward implementation of ZipAR as a baseline on Cosmos, which means we only leverage spatial-level acceleration. Our experimental setup is consistent with the results shown in Table 4. The experimental results are as follows (ws stands for window size, which is a hyperparameter in ZipAR):

Table 4: Comparison between ZipAR (He et al., 2024) and DiagD.

| Model | Algorithm | FVD↓ | Subject Cons.↑ | Image. Qual.↑ | Dynamic degree ↑ | FPS↑ | STEP $(k)$↓ |
|-------|-----------|------|----------------|---------------|------------------|------|-------------|
| 4B | NTP | 136 | **0.978** | **0.604** | 0.49 | 0.38 | 7.68 |
| | ZipAR $ws = 1$ | 349 | 0.823 | 0.379 | 0.53 | 3.42 | 0.30 |
| | DiagD $k = 2$ $d = 10$ | 137 | **0.978** | 0.599 | 0.46 | **3.41** | 0.15 |
| 12B | NTP | **135** | **0.978** | 0.602 | **0.54** | 0.15 | 7.68 |
| | ZipAR $ws = 1$ | 137 | 0.974 | 0.569 | 0.49 | 1.04 | 0.30 |
| | DiagD $k = 1$ $d = 1$ | 139 | 0.967 | 0.564 | 0.51 | 1.71 | **0.11** |

Two interesting findings further distinguish the superiority of Diagonal Decoding in video acceleration compared to the standard ZipAR approach.

- In the 4B model, by balancing spatial and temporal redundancy, DiagD are able to configure parameters such that the number of forward steps is much less than the ZipAR window size of 1 (which corresponds to the maximum acceleration ratio in ZipAR), while still achieving much better generation quality. In contrast, the ZipAR implementation shows severe image degradation under these conditions.

- In the 12B model, by fully exploiting temporal redundancy, we achieve a higher acceleration ratio than ZipAR(1.71 fps v.s. 1.04 fps) at comparable quality levels.

Our motivation incorporates both temporal(inter-frame) and spatial(intra-frame) redundancy (while ZipAR focuses only on spatial redundancy). For video tasks, temporal redundancy is clearly very important. By adjusting two hyperparameters, we can effectively balance spatial and temporal redundancy, achieving higher acceleration and better performance. Our analysis of the model's attention maps also supports this distinction, revealing that temporal redundancy plays a crucial role in accelerating video generation.

## D    DETAILED EXPLANATION OF ALGORITHM

In this section, we clarify our algorithms with examples and more detailed explanation.

### D.1    DIAGONAL DECODING ALGORITHM

In the next token prediction algorithm, the Transformer receives one input token and produces the probabilities for the next token ($[t_1]ß[t_1, t_2]ß[t_1, t_2, t_3]$, .... However, this is just a conventional way of thinking; the Transformer is not limited to receiving only one token at a time. Consider the training phase of the Transformer, where it is actually fed a sequence of n tokens and learns to predict the probability of the next token for each token in the sequence. Therefore, moving from single-token generation to multi-token generation is not inherently difficult—in fact, this transition directly addresses the generation problem illustrated in the figure below. This explains how diagonal generation is performed.

$$\begin{bmatrix} t_1 & t_2 & \cdot \\ t_3 & \cdot & \cdot \\ \cdot & \cdot & \cdot \end{bmatrix} \implies \begin{bmatrix} t_1 & t_2 & t_3 \\ t_4 & t_5 & \cdot \\ \cdot & \cdot & \cdot \end{bmatrix}$$

Next, we discuss how to generate tokens correctly. The content of the tokens to be generated is actually controlled by the position embedding. In a Transformer model, each input token is assigned a position embedding to indicate its position in the sequence. The token to be generated corresponds to the next position for each input token. Therefore, by adding different position embeddings to the input token, we can predict the next token at different positions. For example, to predict $t_5$ , we simply use $t_4$ along with the position embedding corresponding to $t_5$.

Finally, we need to address how to generate the first token of each row, since the last token of the previous row has not yet been generated. By leveraging the spatiotemporal redundancy described in the paper, we select the preceding token as the input. Specifically, to predict $t_7$, we use token $t_4$ and the position embedding corresponding to $t_6$.

### D.2 FINE-TUNING ALGORITHM

The purpose of the attention mask is to address the inconsistency between visible tokens during training and inference. For example, in the case of $t_7$ shown above, causal attention masks in autoregressive training allow $t_7$ to attend to all tokens from $t_1$ to $t_6$. However, in diagonal decoding during inference, $t_6$ has not yet been generated when $t_7$ is produced. This creates some inconsistency, so the newly proposed attention mask ensures that $t_7$ cannot see $t_6$ during training either. Therefore, we only need to modify the traditional causal attention mask during Transformer training to achieve this, and finetuning can be performed in parallel. It is important to emphasize that, unless otherwise stated in the paper, all test results were obtained without finetuning.

## E COSMOS

### E.1 RESULTS ON ROBOTIC MANIPULATION AND DRIVING

Open-ended scenarios indeed involve more complex motion dynamics, but there are few open-sourced autoregressive models performs well on open-ended data. We test DiagD on two more challenging datasets in Cosmos to demonstrate the generalization ability of Diagonal Decoding to open-ended scenarios: the Bridges (Walke et al., 2023) dataset, which focuses on robotic manipulation, and the BDD100K (Yu et al., 2020) dataset, which encompasses diverse driving environments. Both datasets feature highly significant, unpredictable, and pronounced motion as well as complex environments. We randomly selected 100 videos from the test set and randomly extracted 33 continuous frames from each video (no downsampling or frame interpolation was performed). We obtained the following results, Table 5 and 6. It is worth noting that the Cosmos model we used has not been post-trained on robotic or autonomous driving datasets, which limited the capabilities of both next-token prediction and Diagonal Decoding.

Table 5: Quantitative evaluation of Cosmos on BridgeV2 (Walke et al., 2023). "NTP" refers to the next-token prediction paradigm. DiagD $k = i$ $d = j$ denotes the Diagonal Decoding algorithm with different hyper-parameters. "STEP" refers to the number of forward passes required by the model to generate a video.

| Model | Algorithm | FVD↓ | Subject Cons.↑ | Image. Qual.↑ | Dynamic degree | FPS↑ | STEP $(k)$↓ |
|---|---|---|---|---|---|---|---|
| 4B | NTP | 602 | 0.926 | 0.594 | 0.86 | 0.38 | 7.68 |
| | DiagD $k = 4$ $d = 20$ | 615 | 0.928 | 0.592 | 0.87 | 2.66 | 0.26 |
| | DiagD $k = 2$ $d = 40$ | 625 | 0.921 | 0.585 | 0.65 | 3.02 | 0.30 |
| | DiagD $k = 2$ $d = 10$ | 626 | 0.921 | 0.580 | 0.69 | **3.41** | 0.15 |
| 12B | NTP | **585** | 0.926 | 0.590 | 0.80 | 0.15 | 7.68 |
| | DiagD $k = 2$ $d = 40$ | 599 | **0.930** | 0.597 | 0.70 | 1.21 | 0.30 |
| | DiagD $k = 2$ $d = 10$ | 608 | **0.930** | **0.598** | 0.76 | 1.50 | 0.19 |
| | DiagD $k = 1$ $d = 1$ | 609 | 0.923 | 0.590 | 0.69 | 1.71 | **0.11** |

### E.2 DETAILS OF COSMOS MODELS

Cosmos(Agarwal et al., 2025), a World Foundation Model (WFM) Platform for developing Physical AI systems, integrates multiple pre-trained models, including autoregressive and diffusion-based methods, as well as discrete and continuous tokenizers. Specifically, the autoregressive model employs a discrete video tokenizer that leverages a codebook containing $16,000$ entries, achieving spatial compression of $16\times$ and temporal compression of $8\times$. This tokenizer is capable of compressing a video of 33 frames at a resolution of $640 \times 1024$ into $12,800$ discrete tokens. In our study, We implements the spatial and temporal diagonal decoding algorithm on Cosmos autoregressive-based world foundation models. We provided an initial sequence of $5,120$ tokens (equivalent to 9 frames),

Table 6: Quantitative evaluation of Cosmos on BDD100k (Yu et al., 2020). "NTP" refers to the next-token prediction paradigm. DiagD $k = i\ d = j$ denotes the Diagonal Decoding algorithm with different hyper-parameters. "STEP" refers to the number of forward passes required by the model to generate a video.

| Model | Algorithm | FVD↓ | Subject Cons.↑ | Image. Qual.↑ | Dynamic degree | FPS↑ | STEP $(k)$↓ |
|-------|-----------|------|----------------|---------------|----------------|------|-------------|
| 4B | NTP | **567** | 0.946 | 0.500 | 0.99 | 0.38 | 7.68 |
| | DiagD $k = 4\ d = 20$ | 575 | **0.947** | **0.501** | 0.98 | 2.66 | 0.26 |
| | DiagD $k = 2\ d = 40$ | 579 | 0.945 | 0.498 | 0.97 | 3.02 | 0.30 |
| | DiagD $k = 2\ d = 10$ | 585 | 0.940 | 0.492 | 0.99 | **3.41** | 0.15 |
| 12B | NTP | 569 | 0.946 | **0.501** | 0.99 | 0.15 | 7.68 |
| | DiagD $k = 2\ d = 40$ | 568 | **0.947** | 0.499 | 0.99 | 1.21 | 0.30 |
| | DiagD $k = 2\ d = 10$ | 568 | 0.943 | 0.494 | 0.99 | 1.50 | 0.19 |
| | DiagD $k = 1\ d = 1$ | 581 | 0.926 | 0.462 | 0.99 | 1.71 | **0.11** |

optionally accompanied by text depending on the task, to evaluate text-guided video generation and video continuation tasks. This initial sequence was used to generate the subsequent $7,680$ tokens, extending the remaining frames to reach the 33-frame length.

Due to the lack of an open-source evaluation pipeline and datasets, we replicated a comparable setup based on details provided in its technical report. Specifically, we selected 100 videos, each comprising 33 frames, randomly sampled from the RealEstate10K dataset (Zhou et al., 2018). To quantitatively assess visual quality, we employed standard metrics, including Fréchet Video Distance (FVD)(Unterthiner et al., 2018) and Subject Consistency, Dynamic Degree, and Image Quality from VBench (Huang et al., 2024). Furthermore, we conducted a human evaluation, detailed in Section 4.3, to compare visual quality and object movement between videos generated by next-token prediction and DiagD. Unlike Cosmos, our evaluation metrics exclude the use of diffusion decoder for post-processing videos generated by autoregressive models, as this would not fairly reflect the visual quality.

### E.3 Extra Experiments on Hyper-parameters

We report more combination of $k$ and $d$ in Table 7. The FPS values show minimal variation across some settings. This is because the implementation of the diagonal decoding algorithm introduces a small amount of overhead. When the speedup ratio is significantly high, the time lost due to this overhead becomes non-negligible.

### E.4 More Cases

We randomly choose ten cases from 100 evaluation sets in supplementary material.

## F WHAM

### F.1 Details of WHAM

The World and Human Action Model (WHAM) (Kanervisto et al., 2025) is a recently proposed state-of-the-art autoregressive generative model trained on gameplay data from *Bleeding Edge*, capable of generating coherent and diverse gameplay sequences based on user instructions. Unlike Cosmos, WHAM employs an image-level Vector Quantized (VQ) tokenizer that concentrates exclusively on spatial compression. This tokenizer independently converts each game state, with a resolution of $180 \times 300$, into $540$ discrete tokens, which are subsequently concatenated with their corresponding in-game actions. To preserve the inherent relationship between actions and game states, our approach employs spatial diagonal decoding algorithm alone instead of processing the entire video simultaneously. That is to say, we sequentially generate subsequent game states from previous states and their associated actions, alternating between state generation and action concatenation.

For WHAM, we randomly selected 100 videos from its evaluation set to assess video consistency according to WHAM's evaluation protocol. The generation of each video was conditioned on one second of gameplay, which included both video and controller actions, and then proceeded to be

Table 7: Quantitative evaluation on Cosmos. 4B and 12B refer to models used for video continuation. "NTP" refers to the next-token prediction paradigm. $DiagD\ d = m\ k = n$ denotes the Diagonal Decoding algorithm where $d = m$ and $k = n$. "Step" refers to the number of forward passes required by the model to generate a video. "TP" is the number of tokens that model can generate per second.

| Model | Algorithm | FVD↓ | Subject Cons.↑ | Image. Qual.↑ | Dynamic degree ↑ | FPS↑ | TP↑ | STEP $(k)$↓ |
|---|---|---|---|---|---|---|---|---|
| | $d = 2, k = 1$ | 348 | 0.844 | 0.393 | 0.52 | **3.42** | **1280** | **0.11** |
| | $d = 3, k = 1$ | 350 | 0.846 | 0.397 | 0.54 | 3.41 | 1097 | 0.11 |
| | $d = 5, k = 1$ | 352 | 0.859 | 0.402 | 0.51 | 3.41 | 1097 | 0.11 |
| | $d = 9, k = 1$ | 342 | 0.863 | 0.408 | 0.52 | 3.41 | 1097 | 0.12 |
| 4B | $d = 4, k = 2$ | 171 | 0.936 | 0.515 | **0.55** | 3.41 | 1097 | 0.15 |
| | $d = 6, k = 2$ | 145 | 0.966 | 0.574 | 0.49 | 3.41 | 1097 | 0.15 |
| | $d = 10, k = 2$ | 137 | 0.978 | 0.599 | 0.46 | 3.41 | 1097 | 0.16 |
| | $d = 18, k = 2$ | **136** | **0.979** | 0.601 | 0.48 | 3.00 | 960 | 0.24 |
| | $d = 8, k = 4$ | 154 | 0.959 | 0.552 | 0.54 | 2.67 | 853 | 0.24 |
| | $d = 12, k = 4$ | 139 | 0.976 | 0.595 | 0.49 | 2.67 | 853 | 0.24 |
| | $d = 20, k = 4$ | **136** | **0.979** | **0.604** | 0.50 | 2.66 | 852 | 0.26 |
| | $d = 36, k = 4$ | **136** | **0.979** | 0.603 | 0.48 | 2.40 | 768 | 0.29 |
| | $d = 2, k = 1$ | 139 | 0.967 | 0.564 | 0.51 | **1.71** | **549** | **0.11** |
| | $d = 3, k = 1$ | 152 | 0.955 | 0.546 | 0.53 | 1.71 | 549 | 0.11 |
| | $d = 5, k = 1$ | 143 | 0.970 | 0.576 | 0.49 | 1.71 | 549 | 0.11 |
| | $d = 9, k = 1$ | **136** | 0.863 | 0.408 | **0.52** | 1.61 | 515 | 0.12 |
| 12B | $d = 4, k = 2$ | 152 | 0.968 | 0.563 | **0.52** | 1.60 | 512 | 0.15 |
| | $d = 6, k = 2$ | 143 | 0.973 | 0.585 | 0.50 | 1.60 | 512 | 0.15 |
| | $d = 10, k = 2$ | 143 | 0.978 | 0.600 | 0.51 | 1.60 | 512 | 0.16 |
| | $d = 18, k = 2$ | **136** | **0.979** | 0.601 | 0.48 | 1.50 | 480 | 0.18 |
| | $d = 8, k = 4$ | 150 | 0.970 | 0.570 | 0.59 | 1.26 | 427 | 0.24 |
| | $d = 12, k = 4$ | 140 | 0.975 | 0.594 | 0.50 | 1.26 | 404 | 0.24 |
| | $d = 20, k = 4$ | 137 | 0.978 | 0.600 | 0.49 | 1.20 | 384 | 0.26 |
| | $d = 36, k = 4$ | **136** | **0.979** | **0.603** | 0.48 | 1.09 | 349 | 0.29 |

conditioned on the controller actions performed by a human player during the subsequent 10 seconds of gameplay. In addition to reporting Fréchet Video Distance (FVD) and Subject Consistency, Dynamic Degree, and Image Quality from VBench (Huang et al., 2024). Following WHAM's protocol, we conducted a human evaluation to assess the visual quality of generated gameplay and object motion, comparing our results with those obtained using the next-token prediction algorithm.

### F.2 MORE CASES

We randomly choose ten cases of 10 senconds videos from 100 evaluation sets in supplementary material.

## G MC-AR

### G.1 DETAILS OF MC-AR MODELS

We conducted a series of experiments by training models from scratch on the VPT dataset (Baker et al., 2022). The VPT dataset is a domain-specific dataset comprising gameplay videos from *Minecraft*. We employed a pre-trained image VQ-VAE (Patil et al., 2024), an image-level tokenizer with a codebook containing $8,192$ entries, achieving a spatial compression ratio of $16\times$. To enhance visual quality, we subsequently fine-tuned the VQ-VAE on the VPT dataset. Our Transformer model was based on the LLaMA architecture (Touvron et al., 2023) and augmented with 3D Rotary Embeddings (Su et al., 2024). We combine each game state tokens with the corresponding actions just like WHAM, so for each pair of game state and actions in the original input $(x_i, a_i)$, the tokenizers will transfer them into a flat sequence of discrete ids as:

$$(t_{i*c+1}, \cdots, t_{(i+1)*c}, t_1^{a_i}, \cdots, t_n^{a_i}). \tag{7}$$

and $c$ is the number of ids to represent each state, $n$ is the number of actions. We trained our model on next token prediction tasks, enabling the model to predict future states based on previous game states and current action. We use the Adam optimizer(Kingma & Ba, 2015) with a cosine decay learning rate scheduler to train the model. Additionally, fine-tuning was performed for an extra $1,000$ steps on the same dataset.

For MC-AR, we selected 100 video clips from an unused subset of the evaluation set, each containing 16 frames, with the last 15 frames corresponding to actions in the gameplay. Each model generated 15 subsequent frames conditioned on the first frame of each clip and the 15 actions. These generated frames were then compared against the ground truth using the FVD, Subject Consistency, Dynamic Degree, and Image Quality metrics.

### G.2 MODEL CONFIGURATIONS

We train three different sizes of the model within the LLaMA architecture: 300M, 700M, and 1.2B. We tune the hidden dimension, intermediate dimension, and the number of layers to achieve different model sizes. The configuration of these models are listed in Table 8. The hyperparameters of the optimizer used to train the model are listed in Table 9.

Table 8: The configuration of different size of models.

|  | Hidden dim | MLP dim | Num. Heads | Num. Layers |
|---|---|---|---|---|
| 300M | 1024 | 4096 | 16 | 20 |
| 700M | 2048 | 4096 | 32 | 20 |
| 1.2B | 2048 | 8192 | 32 | 20 |

Table 9: Optimization hyperparameters.

| Hyperparameter | Value |
|---|---|
| Learning rate scheduler | cosine |
| Learning rate | $3e^{-4}$ |
| Warm up steps | 10000 |
| Weight decay | 0.1 |
| Optimizer | AdamW |
| AdamW betas | $(0.9, 0.95)$ |
| Maximum Positions | 5376 |

Table 10: Quantitative evaluation on 300M MC-AR. We use DiagD ($k = 2$) in experiment.

| Algorithm | FVD↓ | Subject Cons.↑ | Image. Qual.↑ | Dynamic degree ↑ | FPS↑ | STEP ($k$)↓ |
|---|---|---|---|---|---|---|
| NTP | **223** | **0.869** | **0.676** | 0.98 | 1.08 | 5.04 |
| DiagD $k = 2$ w/o FT | 246 | 0.854 | 0.650 | **0.99** | **3.98** | **0.75** |
| DiagD $k = 2$ w/ FT | 233 | 0.845 | 0.648 | 0.98 | **3.98** | **0.75** |

### G.3 EXTRA EXPERIMENTS

We provide models of three scales (300M Table 10, and 1.2B Table 11) to present additional results on MC-AR.

### G.4 MORE CASES

We randomly choose ten cases from 100 evaluation sets in supplementary material.

Table 11: Quantitative evaluation on 1.2B MC-AR. We use DiagD ($k = 2, 4$) in experiment.

| Algorithm | FVD↓ | Subject Cons.↑ | Image. Qual.↑ | Dynamic degree ↑ | FPS↑ | STEP ($k$)↓ |
|---|---|---|---|---|---|---|
| NTP | **203** | **0.866** | **0.677** | 0.97 | 0.89 | 5.04 |
| DiagD $k = 4$ w/o FT | 246 | 0.857 | 0.645 | **0.98** | 1.42 | 1.14 |
| DiagD $k = 2$ w/o FT | 246 | 0.841 | 0.606 | 0.97 | **1.98** | **0.75** |
| DiagD $k = 2$ w/ FT | 227 | 0.853 | 0.661 | **0.98** | **1.98** | **0.75** |

## H  DERIVATIONS

We derive Equation (4) and (6) here. First, for Equation (4), assume $\min\{h, w\} = h$, we have:

$$
\begin{aligned}
r_{\text{spa}} &= \frac{h \cdot w}{(h - 1) \cdot k + w} \\
&= \frac{h}{\frac{h}{w} \cdot k - \frac{k}{w} + 1} \\
&\approx \frac{h}{\frac{h}{w} \cdot k + 1}.
\end{aligned}
\tag{8}
$$

Where we assume $\frac{k}{w} \approx 0$ which is applicable for most of our cases. And as a result, the approximation in Equation (4) achieves if $h \approx w$.

Similarly, for Equation (6), we have:

$$
\begin{aligned}
r_{\text{diag}} &= \frac{T \cdot h \cdot w}{(T - 1) \cdot h + h + w - 1} \\
&= \frac{T \cdot h \cdot w}{T \cdot h + w - 1} \\
&\approx \frac{w}{1 + \frac{w}{T \cdot h}} \\
&\approx w
\end{aligned}
\tag{9}
$$

Where the approximation stands when $T * h \gg w$, which is applicable for most of video generation cases.

