# OpenReview forum: "Fast Autoregressive Video Generation with Diagonal Decoding"
_ICLR.cc/2026/Conference — ICLR 2026 Conference Withdrawn Submission_

### Official Review · Reviewer_4G3v · 2025-10-31

**Soundness:** 3
**Presentation:** 3
**Contribution:** 3
**Rating:** 4
**Confidence:** 4

**Summary:**

This paper proposes a **Diagonal Decoding** scheme for autoregressive (AR) video generation. The method enables the model to predict multiple tokens simultaneously, significantly improving video generation speed without sacrificing generation quality.

**Strengths:**

- The paper is well-organized and clearly written, with a strong logical flow and clear presentation of the proposed method.
- The proposed approach effectively exploits both intra-frame local redundancy and inter-frame temporal redundancy, achieving substantial acceleration in video generation.

**Weaknesses:**

1. Compared with related work such as **ZipAR**, the proposed method appears to be a straightforward extension of ZipAR from images to videos, and the novelty may be somewhat limited. Although the authors claim that their method achieves higher acceleration in the video domain (Line 128), this comparison seems **unfair**, since videos naturally contain more redundancy than images, leaving greater room for speedup. Comparing video acceleration with image acceleration under this setting may not be meaningful.
2. The conclusions in **Table 2** and **Table 3** are not entirely consistent. Table 2 shows that fine-tuning DiagD slightly reduces or maintains the Dynamic Degree, while Table 3 indicates a significant improvement after fine-tuning. This discrepancy raises the question of whether the observed performance gain originates from the fine-tuning dataset itself rather than from the proposed DiagD method.

**Questions:**

See weakness above

---

### Official Review · Reviewer_FFfj · 2025-10-31

**Soundness:** 4
**Presentation:** 3
**Contribution:** 3
**Rating:** 8
**Confidence:** 3

**Summary:**

This work proposes Diagonal Decoding (DiagD), a training-free inference acceleration method for autoregressive video generation models. The key strength lies in its simple yet effective insight: leveraging spatial and temporal correlations in videos by generating tokens along diagonal paths, enabling parallel decoding within and across frames. Despite DiagD being similar to ZipAR in spatial correlation parallel decoding, its insight into temporal correlation highlights the novelty of this work.

This versatile method can be applied to various model architectures and tasks, achieving a speed increase of up to 10× with slight quality degradation. This has been demonstrated across multiple models and datasets. Extensive evaluation, including human studies, provides strong empirical support for the method's effectiveness and practicality. However, DiagD shows a reduction in video dynamic degree in larger-scale video generative models. I hope to see more discussion and study on this topic. In addition, this work proposes improving performance by fine-tuning to mitigate the training and inference gap. I would like to see a more detailed explanation of this.

Overall, I like this work and advocate for acceptance.

**Strengths:**

1. This work proposes Diagonal Decoding (DiagD), a training-free inference acceleration method for autoregressive video generation models.
2. The key strength lies in its simple yet effective insight: leveraging spatial and temporal correlations in videos by generating tokens along diagonal paths, enabling parallel decoding within and across frames. Despite DiagD being similar to ZipAR in spatial correlation parallel decoding, its insight into temporal correlation highlights the novelty of this work.
3. This versatile method can be applied to various model architectures and tasks, achieving a speed increase of up to 10× with slight quality degradation. This has been demonstrated across multiple models and datasets.
4. This work also proposes to mitigate the gap between training and inference through finetuning, and prove this on MC-AR task (game scenes and actions on the game Minecraft).

**Weaknesses:**

1. DiagD shows a reduction in the dynamic degree of video in larger-scale generative models. I hope to see more discussion and research on this topic. Intuitively, such temporal-based parallel decoding will lose temporal information, thus reducing the dynamic degree.
2. In addition, this work proposes improving performance through fine-tuning to mitigate the training and inference gap. However, this has only been proven experimentally in the MC-AR task, which is not convincing enough.
3. Regarding the claimed scaling effects in Section 4.3, I would expect to see a more comprehensive analysis or visualization to provide support.

**Questions:**

1. Is the reduction of the dynamic degree an inevitable problem of parallel decoding?
2. Will there be a more comprehensive introduction about the finetune optimization to mitigate the gap between training and inference?

---

### Official Review · Reviewer_1zzu · 2025-11-01

**Soundness:** 3
**Presentation:** 3
**Contribution:** 2
**Rating:** 2
**Confidence:** 4

**Summary:**

This paper proposes Diagonal Decoding (DiagD) to accelerate autoregressive (AR) video generation by exploiting spatial–temporal redundancy. Instead of raster-scan next-token prediction, DiagD generates tokens along spatiotemporal diagonals, enabling parallel decoding within a frame and partial overlap across frames. The method is largely training-free and can be optionally fine-tuned by replacing the causal mask with a DiagD-aligned attention mask. Experiments on Cosmos, WHAM, and a Minecraft-style AR model report up to 10× speedups with small quality drops, plus minor fine-tuning to close the gap.

**Strengths:**

1. Clear, practical acceleration idea. Change decoding to diagonal and parallel is intuitive.
2. Training-free with an optional, lightweight fine-tune. The attention-mask swap is a minimal modification that addresses the training–inference ordering gap without major retraining.
3. Strong empirical speedups. Reported wall-clock and step reductions are substantial, with competitive FVD/VBench and human preferences under moderate acceleration settings.

**Weaknesses:**

1. Limited novelty. I hate to say this but the idea is a simple extension of image-based acceleration instead of a fundamentally new modeling principle; related works (spatial-parallel decoders, speculative/Jacobi decoding, random-order AR) reduce sequential dependence in similar manner.
2. The independence assumptions underlying diagonal parallelism are plausible but not quantified. There is no analysis of error propagation or sufficient conditions under which diagonal groups are safe to decode jointly.
3. External validity is unclear. Evaluation focuses on controlled environments (world models, games) and a limited open-domain proxy; robustness under complex, non-redundant motion and heavy occlusions is not convincingly stress-tested. As an acceleration method, generality is very important.
4. In Tab.  2, the gap between NTP and DiagD is huge. The optional fine-tuning/mask realignment is key to closing quality gaps, so the zero-training claim is somewhat overstated.

**Questions:**

Please refer to the weaknesses.

---

### Official Review · Reviewer_9SGt · 2025-11-01

**Soundness:** 2
**Presentation:** 2
**Contribution:** 3
**Rating:** 4
**Confidence:** 4

**Summary:**

This paper observes that video frames have strong local spatial correlation compared to distant locations, and thus propose to perform parallel frame decoding following the diagonal trajectory. The proposed method can be applied on pre-trained AR video models, and addresses the time bottleneck of their original sequential decoding paradigm. It can be adapted training-free or with finetuning. The proposed method is tested on multiple video frame prediction benchmarks, achieving large speed-up with minimal quality degradation.

**Strengths:**

- This work has good motivation and observation that video content is spatially locally correlated across frames, and thus proposes to decode only dependent on the spatially local content of previous frames, i.e. from a corner and following diagonals.

- The proposed method is (almost) training-free and can be handily adapted to pre-trained AR models to gain big speed-up with minimal quality drop.

**Weaknesses:**

- The proposed method is only evaluated on frame prediction task, where a domain-specific model continue the video given a leading clip. Not only the task complexity but also the core observation might depend on the video types, e.g. egocentric videos vs action videos. It would be further highlighted to test with open-domain T2V AR models on general video generation only conditioned on non-visual input, especially on general videos like human actions, object moving with fixed camera view, etc. to solidate the observation and proposed method's generability.

- The quantitative metrics need to be upgrade. FVD is highly sensitive to statistic data and thus needs to be calculated on at least thousands of samples to be stable enough. For general open-domain T2V cases, more comprehensive benchmarks like VBench etc. should be involved.

**Questions:**

- How the VRAM will change when applied the parallel diagonal decoding? This is not a major issue, but would help comprehensive understanding the full trade-off.

---

### Note · Authors · 2025-11-14

I have read and agree with the venue's withdrawal policy on behalf of myself and my co-authors.